# Fast Anchor Point Matching for Emergency UAV Image Stitching Using Position and Pose Information

**DOI:** 10.3390/s20072007

**Published:** 2020-04-03

**Authors:** Ruizhe Shao, Chun Du, Hao Chen, Jun Li

**Affiliations:** Department of Cognitive Communication, College of Electronic Science and Technology, National University of Defense Technology, Changsha 410000, Hunan, China; shaoruizhe@nudt.edu.cn (R.S.); duchun@nudt.edu.cn (C.D.); junli@nudt.edu.cn (J.L.)

**Keywords:** anchor points, UAV, image stitching, feature points

## Abstract

With the development of unmanned aerial vehicle (UAV) techniques, UAV images are becoming more widely used. However, as an essential step of UAV image application, the computation of stitching remains time intensive, especially for emergency applications. Addressing this issue, we propose a novel approach to use the position and pose information of UAV images to speed up the process of image stitching, called FUIS (fast UAV image stitching). This stitches images by feature points. However, unlike traditional approaches, our approach rapidly finds several anchor-matches instead of a lot of feature matches to stitch the image. Firstly, from a large number of feature points, we design a method to select a small number of them that are more helpful for stitching as anchor points. Then, a method is proposed to more quickly and accurately match these anchor points, using position and pose information. Experiments show that our method significantly reduces the time consumption compared with the-state-of-art approaches with accuracy guaranteed.

## 1. Introduction

With the development of unmanned aerial vehicle (UAV) techniques, aerial images are becoming cheaper, easily accessed, and of higher resolution. In many fields, such as surveying, mapping, resource exploration, and disaster monitoring, remote-sensing images from UAVs are widely used.

In some emergency situations, such as disaster rescue, image-stitching results need to be obtained rapidly. If the computational complexity of the stitching algorithm is low enough, a portable computation device (for example, a laptop or tablet PC) can generate the real-time stitched images of the target area on site after the UAV is released for investigation. Such a process can support the first aid teams in disaster rescue actions, such as earthquakes, floods, mud-rock flow or avalanches. Therefore, reducing the computation cost of UAV image stitching holds great promise for meaningful outcomes.

Works in the literature have proposed various approaches to stitch UAV images, however, most of them address the improvement of precision. Some approaches [1,2,3] are presented to improve the stitching speed. However, the stitching time of these methods is still too long to meet the needs of emergency applications.

Utilizing the Global Positioning System (GPS) and inertial measurement unit (IMU) carried by the UAV, the position and pose information of the UAV image can be obtained easily. The position and pose information can determine the approximate position of the UAV image. However, limited by the accuracy of the instrument, the accuracy of the position information cannot meet the requirements for stitching.

As described above, conventional image-stitching-based methods face the disadvantage of slow speed, while the position and pose information of the image recorded by UAV do not have sufficient accuracy for stitching. To address this problem, this paper proposes a stitching approach that uses some optimization methods to simplify the stitching computation with position and pose information. This approach stitches images by finding several anchor-matches instead of a large number of feature-matches, and reducing the range where features need to be extracted and the number of feature points that need to be matched, which is why it is faster.

The main contributions of this paper are as follows:A novel UAV image-stitching approach based on feature point matching is proposed. Compared to mainstream methods [4,5], it can use position and pose information effectively and reduce the computation time significantly. The main purpose of our approach is to fast stitch UAV images during emergency situations, represented by its name, FUIS (fast UAV image stitching).A novel anchor points selection approach is designed, which can select fewer anchor points from a large number of feature points to accelerate the stitching process with accuracy guaranteed.To validate the proposed approach, we conducted experiments to compare FUIS to existing approaches. The result shows that FUIS can be faster than the existing approaches with guaranteed accuracy.

The remainder this paper is organized as follows. In Section 2, related work on UAV image stitching is introduced. The problem of UAV image stitching is clearly defined in Section 3. Then, Section 4 introduces FUIS by first outlining and then detailing the main points. In Section 5, the experiments and discussions are presented. Finally, the conclusions are drawn in Section 6.

## 2. Related Works

UAV image stitching has a lot in common with general image stitching. It refers to the technique that merges an obtained sequence of two or more UAV images into one wide-field image by finding an appropriative image transformation model [6]. Various methods are proposed to find the model, such as region-based methods, represented by phase correlation [7], however, feature-point-based methods are more widely used [8].

The feature-point-based methods align images by finding the common feature points of the two images. They are mainly divided into three steps: feature point extraction and description, feature point matching, and image transformation. For feature extraction, the mainstream methods are SIFT (scale-invariant feature transform) [9], SURF (speeded up robust features) [10,11], KAZE [12], and ORB (oriented fast and rotated brief) [13]. In addition to the handmade feature points, deep learning methods, like LIFT (Learned Invariant Feature Transform) [14], are also applied in this field. These feature algorithms are widely used in various applications of UAV image stitching [3,5,15,16,17]. For feature matching, some excellent algorithms are applied, such as RANSAC (random sample consensus) [18], GMS (grid-based motion statistics) [19,20]. These methods improve the accuracy of feature points matching, and avoid the adverse effect of error-matched points on aligning. Therefore, some UAV image-stitching approaches adopt these methods for better performance [19,21]. Then, the matched feature pairs are used to calculate the transformation model to align images. In this step, the transformation homography matrix is widely used because it is simple and effective, as UAV image stitching mainly addresses stitching planar scenes [22]. However, to obtain more accurate results, some literatures [17,23,24,25,26] proposed more complex models and effectively removed parallax.

After finding the appropriative image transformation model, image fusion is usually performed. The main purpose of fusion is to enhance the visual effect and remove the seam line effect. Currently, image-blending methods [27,28] and seam line extraction methods [29,30,31] are commonly used because of their excellent performance.

In addition to traditional image-stitching methods, some methods that are specially designed for UAV image stitching are proposed. Compared to general image stitching, UAV image stitching has unique characteristics. First, as an aerial image, the scene is the ground which is approximately planar, and the subject distance, which is the height, does not change drastically. Second, the position and pose information of the UAV when it took the image is available, but because a UAV is too small, the stability and accuracy of this information is low. Using these characteristics, the literature has proposed some optimization approaches to improve the accuracy and speed of the stitching process. Camera position information is used by [1] to streamline images by removing unnecessary frames, and reducing the cumulative error by finding the lateral relative positional relationship of images and matching these images. Li et al. [2] assumes the height of the UAV is constant, and reduces the computation by removing some octaves from SIFT. GPS is used by [32] to predict the position of the feature points to be extracted in the next key frame, thereby reduce the time consumption of stitching. However, this approach does not use the pose information and still consumes a lot of time to process a large number of feature points. These approaches reduce the computational cost to some extent, but they do not make full use of the position and pose information recorded by a UAV, and are still too time-consuming to meet the needs of some emergency applications. That is the problem we have addressed.

## 3. Problem Definition

The premise of the problem is as follows. The UAV images of the target area are denoted as $\left\{I_{1},I_{2},\ldots \ldots ,I_{n}\right\}$. These images have the overlapping field of view at a certain overlapping rate. The position information (including altitude, latitude, and longitude), the pose information (including pose angles of the camera ω,φ and κ) of the UAV when it took the images are available, denoted as $\left\{camera_{i}\right\},i=1,2,\ldots ,n,camera_{i}=\left(alt_{i},lat_{i},lng_{i},\omega_{i},\varphi_{i},\kappa_{i}\right)$. The focal length is f. Therefore, we can find transform functions inside the homography domain $\left\{F_{1},F_{2},\ldots \ldots ,F_{n}\right\},F_{i}\in ℍ$ to stitch the images together.

Using the premise above, we define the emergency UAV image-stitching problem as follows:

Given $\left\{camera_{i}\right\}$ and the UAV images $\left\{I_{i}\right\}$, emergency UAV image stitching is to find appropriate transformation functions $\left\{F_{1},F_{2},\ldots \ldots ,F_{n}\right\}$ in the least amount of time, so that $\left\{F_{1},F_{2},\ldots \ldots ,F_{n}\right\}$ can stitch $\left\{I_{1},I_{2},\ldots \ldots ,I_{n}\right\}$ together to produce a combined image I with a certain degree of accuracy.

Ideally, a set of absolutely accurate transformation functions $\left\{F_{1},F_{2},\ldots \ldots ,F_{n}\right\}$, should make the pixels of the same feature in different images $\left\{I_{1},I_{2},\ldots \ldots ,I_{n}\right\}$ be transformed to the same location in I. Assume there are k features $\left\{p_{1},p_{2}\ldots \ldots ,p_{k}\right\}$, each feature appears in $m_{i}\left(i=1,2,\ldots \ldots ,k\right)$ images, then we have to find transformation functions $\left\{F_{1},F_{2},\ldots \ldots ,F_{n}\right\}$ that satisfy:
(1)$$\begin{array}{c} F_{1}\left(x_{1}^{\left(i\right)}\right)=F_{2}\left(x_{2}^{\left(i\right)}\right)=\ldots F_{m_{i}}\left(x_{m_{i}}^{\left(i\right)}\right)\;i=1,2,\ldots \ldots ,k \end{array}$$
where $x_{m}^{\left(i\right)}$ is the pixel coordinate of $p_{i}$ in $I_{m}$. In practice, we should make Equation (1) approximately hold.

When finding the transformation functions, we can use the position and pose information to simplify the calculation or use a parallel computation strategy to accelerate the stitching process.

## 4. Methodology

### 4.1. Overview

To address the problem defined above, we propose a novel stitching approach. The overview of our approach (FUIS) is shown in Figure 1. 

Rough stitching is the preparation for optimized stitching. The principle is as follows.

According to the photogrammetry theory [33], the projection relation between the world coordinate and the camera coordinate can be determined:
(2)$$\begin{array}{c} \frac{Z_{c}}{D}\left[\begin{array}{c} x\\ y\\ 1 \end{array}\right]=K\left[\begin{array}{cc} R & T\\ 0 & 1 \end{array}\right]\left[\begin{array}{c} X_{w}\\ Y_{w}\\ Z_{w}\\ 1 \end{array}\right] \end{array}$$
where $Z_{c}$ is Z value in the camera coordinate. D is the pixel pitch of the imaging sensor and (x,y) is the pixel coordinate on the UAV image. R,T and K represent the rotation matrix, the offset vector and the projection matrix, respectively.

Using the position, pose, and camera parameters, the above parameters can be determined. The image coordinates in image $I_{1}$
$\left(x_{1},y_{1}\right)$ are mapped to world coordinates $\left(X_{w},Y_{w},Z_{w}\right)$, and then mapped to the image coordinates in image $I_{2}$
$\left(x_{2},y_{2}\right)$. In this way, the mapping relationship between two images $I_{1}$ and $I_{2}$ is obtained: $\left(x_{2},y_{2}\right)=f\left(x_{1},y_{1}\right)$, producing the rough registration between the pictures.

The three steps: selecting the anchor points, matching them to obtain anchor point pairs, and calculating the transformation matrix, are the core of image stitching and the focus of our optimization. Feature point matching will inevitably have errors, including error-matches and matching deviations. The traditional methods guarantee robustness against these errors by increasing the number of feature matches. They extract and match many more feature points than needed to make sure the incorrect matches are a minority. And then filter out the incorrect matches using some methods, such as RANSAC. However, the initial inclusion of large numbers of feature points also requires extensive calculations. To overcome this difficulty, we select anchor points from feature points and propose a method to improve the matching accuracy. These methods will be detailed in the following Section 4.2, Section 4.3 and Section 4.4.

For a sequence of images $\left\{I_{0},I_{1},I_{2}\ldots \ldots ,I_{n}\right\}$ taken by a UAV, the adjacent image frames are also adjacent in position. Under this condition, we use the following method to stitch the image sequence. Firstly, according to the above steps, the transformation matrixes between each two adjacent images in the sequence are determined, denoted as $\left\{H_{1},H_{2}\ldots \ldots ,H_{n}\right\}$. Then, the first image is used as the reference image and the other images are transformed to the coordinate of the first image. That is, multiply $\left\{H_{i}\right\}$ in series to obtain the transformation matrixes that transform each image to the stitching result $\left\{H_{0}^{*},H_{1,}^{*}\ldots \ldots ,H_{n}^{*}\right\}$.
(3)$$\begin{array}{c} H_{i}^{*}=\{\begin{array}{c} I,\;\left(i=0\right)\\ H_{i}H_{i-1,}^{*}\left(i=1,2,\ldots ,n\right) \end{array} \end{array}$$

The advantage of this method is that the matching task is easy to divide, and, therefore, it is easy to implement in parallel. This will be proven in the later experimental section.

The seam line technology refers to the technology of selecting the best seam line in the overlap area, aimed at eliminating the stitching traces brought about by the geometric misalignment, and improving the stitching visual effect. The Voronoi graph method [30,31] is one of the seam line extraction methods. The principle of this method is to let the seam line pass through the center of the overlapping area instead of the corner, for the deviation at the center of the overlapping area is generally smaller than that at the edge. This paper applies the Voronoi graph method to extract the seam line since few calculations are required and the stitching effect is significantly improved.

### 4.2. Feature Extraction and Anchor Point Selection

In this section, we extract feature points and select several most stitching-helpful ones as anchor points to be matched in the next step. We call these points anchor points.

#### 4.2.1. Find Feature Points Inside the Overlapped Area

According to the rough registration results, we can use the Sutherland–Hodgman algorithm [34] to find the approximate overlapped area of the adjacent images. In the matching process, only the feature points inside the overlapped area are useful, so we only extract feature points inside the overlapped area. The process is simplified by extracting fewer points, as compared to the traditional full picture extraction.

#### 4.2.2. Use Speeded up Robust Features (SURF) as Feature Extractor and Descriptor

There are many excellent feature-extraction algorithms to choose from, including ORB, SURF, and SIFT. Considering both stitching speed and accuracy, in this paper, we select SURF as the feature descriptor. According to our experiments, compared with ORB, SURF has a higher quality of feature points, which means a higher accuracy of matching. While compared with SIFT, SURF has a faster speed [10,11], due to the use of box filters and Haar wavelet filters. The parameters of SURF will be further discussed in Section 5.1.

#### 4.2.3. Select Anchor Points

To select the most stitching-helpful ones in the extracted feature points as anchor points, we designed the selecting method in accordance with the following principles.

a)Given priority to feature points with large response

The response is the result of the feature extractor acting on the image. Feature points with larger response are more likely to be more remarkable and discernible, and thus, feature matches between these feature points are more likely to be successful. When we select anchor points, priority is given to the feature points with larger response.

b)Select an appropriate number of anchor points

It is obvious that the fewer the anchor points, the faster our algorithm. Since the transformation matrix has 8 degrees of freedom, theoretically four pairs of matches are enough to solve the transform matrix. But if the anchor pairs are few, the stitching error will be more sensitive to the error of each point. We find the best tradeoff through the experiments discussed in Section 5.4.1.

c)Make the distance between the anchor points as large as possible

In this part, we will discuss what principles of spatial distribution should be followed when we select anchor points.

Given a set of anchor point matches $S=\left\{\left(p_{n},q_{n}\right)\right\}$, where $p_{n}$ and $q_{n}$ are the locations of the matching anchor points in the adjacent images $I_{1}$ and $I_{2}$, respectively. Due to various matching errors, the positions of points have noises $\delta_{pn}$ and $\delta_{qn}$. The accurate position is $\left(p_{n}^{*},q_{n}^{*}\right)=\left(p_{n}-\delta_{pn},q_{n}-\delta_{qn}\right)$. Our purpose is to determine an anchor point spatial distribution that can obtain the transformation between two images with the least error when the number of matches and $\delta_{pn}$, $\delta_{qn}$ is constant.

To make the problem more intuitive and less complex, we decompose the image transformation into translation, rotation, and scaling, and analyze them separately. Among them, the translation is unrelated to the distribution of anchor points. Thus, we focus on the rotation and scaling. 

Assume we have two matches $\left(p_{1},q_{1}\right)$ and $\left(p_{2},q_{2}\right)$. The corresponding noise is $\left(\delta_{p1},\delta_{q1}\right)$ and $\left(\delta_{p2},\delta_{q2}\right)$. The accurate position is $\left(p_{1}^{*},q_{1}^{*}\right)$ and $\left(p_{2}^{*},q_{2}^{*}\right)$. Denote $l_{1}=p_{1}-p_{2}$, $l_{2}=q_{1}-q_{2}$ and $\delta_{1}=\delta_{p1}-\delta_{p2}$, $\delta_{2}=\delta_{q1}-\delta_{q2}$. Use the anchor point matches to estimate the rotation angle θ, we have:(4)$$cos\hat{\theta }=\frac{l_{1}\cdot l_{2}}{\left|l_{1}||l_{2}\right|}$$
while true θ is:(5)$$cos\theta =\frac{\left(l_{1}-\delta_{1}\right)\cdot \left(l_{2}-\delta_{2}\right)}{\left|l_{1}-\delta_{1}\right|\left|l_{2}-\delta_{2}\right|}$$

The error of the estimated angle is:(6)$$\Delta cos\hat{\theta }=cos\hat{\theta }-cos\theta =\frac{l_{1}l_{2}}{\left|l_{1}l_{2}\right|}-\frac{\left(l_{1}-\delta_{1}\right)\left(l_{2}-\delta_{2}\right)}{\left|l_{1}-\delta_{1}\right|\left|l_{2}-\delta_{2}\right|}\approx \frac{l_{1}}{\left|l_{1}\right|}\cdot \frac{\delta_{2}}{\left|l_{2}\right|}+\frac{l_{2}}{\left|l_{2}\right|}\cdot \frac{\delta_{1}}{\left|l_{1}\right|}$$

As can be seen from the above equation, for a certain $\delta_{1}$, $\delta_{2}$, if we want to make the error of the estimated angle $\Delta cos\hat{\theta }$ smaller, we need to select anchor points with greater distance between each other, which is $\left|l_{1}\left|\;or\;\right|l_{2}\right|$. 

Similarly, when we estimate the scale factor s, we have:
(7)$$\begin{array}{c} \hat{s}=\frac{\left|l_{1}\right|}{\left|l_{2}\right|} \end{array}$$while true s is:
(8)$$\begin{array}{c} s=\frac{\left|l_{1}-\delta_{1}\right|}{\left|l_{2}-\delta_{2}\right|} \end{array}$$The relative error of the estimated s is:
(9)$$\begin{array}{c} \frac{\Delta s}{\hat{s}}=\frac{\hat{s}-s}{\hat{s}}=1-\frac{\left|l_{1}\right|\left|l_{2}-\delta_{2}\right|}{\left|l_{2}\right|\left|l_{1}-\delta_{1}\right|}\approx 1-\sqrt{1-\frac{2l_{2}\cdot \delta_{2}}{{\left|l_{2}\right|}^{2}}} \end{array}$$

Similar to the angle estimation, to minimize the error of the estimated scale factor Δs, we should still increase the distance between feature points $\left|l_{1}\left|\;or\;\right|l_{2}\right|$. It should be noted that when calculating the transformation model, we directly solve the transformation matrix H instead of calculating multiple transformation factors separately, such as θ and s. However, the conclusion that the greater the distance between anchor points, the smaller the error is consistent. 

From the theoretical derivation above, we find that for a certain match error and a limited number of anchor points, to obtain a transformation model with as little stitching error as possible, we need to make the distance between anchor points ($\left|l_{1}\left|\;and\;\right|l_{2}\right|$) as large as possible. Intuitively, if the anchor points can uniformly distribute throughout the image, there will not be a situation where areas with dense anchor points have small error, while areas with sparse anchors or even no anchor points have large error.

According to all the principles above, we designed the grid method to select the anchor points, so that the distance between them is larger and they are evenly distributed in the whole image. The specific process is: first, use S-H algorithm [34] to find the overlapped area of two adjacent images $I_{1}$ and $I_{2}$. Choose $I_{1}$ as the reference image and determine the minimum enclosing rectangle (MER) of the overlapped area in $I_{1}$. Second, extract SURF feature points in this MER. Third, divide the MER into n*n grids. (The selection of n will be further discussed in experiment Section 5.4.1) Select the feature points that have the largest SURF response from each grid as anchor points, and then find the feature point that matches to it in $I_{2}$. The method to find the matching feature point will be described in Section 4.3.

With this grid selection method, we can guarantee that no more than one anchor point in the same grid will be selected and, thus, the distance between the two anchor points $\left|l_{1}\left|\;and\;\right|l_{2}\right|$ is no less than the distance between the grids in which they are located. Therefore, according to Equations (6) and (9), we can obtain a transformation model with less error.

### 4.3. Find Matching Feature Points in the Neighborhood Window

In this section, we will attempt to ascertain the feature points in $I_{2}$ that match the anchor points, and determine whether they match correctly. If the match is correct, it can be used as an anchor point pair.

The traditional stitching method matches the feature points of the whole images, creating two drawbacks. First, there are too many feature points to participate in the matching, which greatly increases the amount of calculation. Second, the more the feature points, the more similar points, thus increasing the probability of error-match. To address these two problems, we propose our matching method: Firstly, we use the neighborhood window to increase the accuracy and reduce the computation. Secondly, we use the threshold of feature matching to filter the error matches.

#### 4.3.1. Neighborhood Window

Generally, the points in $I_{2}$ that match each anchor point in $I_{1}$ are located near their corresponding positions determined by rough registration. Therefore, we just have to search nearby. Compared with whole picture matching, this matching method has two advantages: first, the amount of required calculations is reduced, resulting in improved stitching speed. Second, in this method, only feature points that are consistent in the feature description and reasonable in position at the same time can be matched. The feature points whose feature descriptors are similar but far from their corresponding positions cannot be matched. This method is equivalent to adding a position constraint to the feature matches in addition to the descriptor. Thus, partial error matches that may occur in whole image matching are avoided.

To facilitate the use of the existing SURF feature point extraction algorithm, the neighborhood adopted in this chapter is a square neighborhood window, which is centered at the corresponding position, with 2×margin as the side length. Assume the rough registration error brought by GPS error is n~N(0,σ) and the SURF template’s size of that feature is t, set the window size margin to:
(10)$$\begin{array}{c} margin=3\sigma +t \end{array}$$

#### 4.3.2. The Scale of the Feature

Since the height of the UAV does not change drastically when taking adjacent images, the scale of the same feature should be similar. Therefore, when we find matches, we only consider those with similar feature scales.

#### 4.3.3. Feature Match Threshold

We use the Euclidean distance Δ of the two descriptors to match the feature and determine whether the match is correct.
(11)$$\begin{array}{c} \Delta =\overset{63}{\underset{i=0}{\sum }}{\left(v_{p}^{i}-v_{q}^{i}\right)}^{2} \end{array}$$where $v_{p}^{i}$ and $v_{q}^{i}$ represent the dimension i of the feature descriptors. For the SURF feature descriptor, the number of descriptor dimensions is 64.

If Δ is less than a threshold (which will be further discussed in Section 5.4.2), then the matching is successful; otherwise the matching fails. This may occur when the error of GPS exceeds our expectation or in an instance of excessive terrain undulation. In this case, this anchor point should be discarded, and the feature point that has the largest response other than this point will try to be matched as the anchor point, and so on. The process repeats until achieving the correct feature match that meets the threshold.

Figure 2 shows the results of feature point matching in two 4000×3000 aerial images $I_{1}$(left) and $I_{2}$(right). The squares in $I_{2}$ are the neighborhood windows around the corresponding area of the feature point. The lines connecting the two images represent pairs of anchor points. The black points in $I_{1}$ are the anchor points selected according to the method of Section 4.2, and the right endpoints of the lines are the feature points that match the anchor points. 

### 4.4. Calculate the Transform Matrix with Added Constraints

We define the homography matrix as:
(12)$$\begin{array}{c} H=\left[\begin{array}{cc} \begin{array}{cc} h_{11} & h_{12}\\ h_{21} & h_{22} \end{array} & \begin{array}{c} h_{13}\\ h_{23} \end{array}\\ \begin{array}{cc} h_{31} & h_{32} \end{array} & 1 \end{array}\right] \end{array}$$

Then we substitute the anchor points matches into the following formula to solve H.


(13)$$\begin{array}{c} w\left[\begin{array}{c} \begin{array}{c} x^{\prime }\\ y^{\prime } \end{array}\\ 1 \end{array}\right]=\left[\begin{array}{cc} \begin{array}{cc} h_{11} & h_{12}\\ h_{21} & h_{22} \end{array} & \begin{array}{c} h_{13}\\ h_{23} \end{array}\\ \begin{array}{cc} h_{31} & h_{32} \end{array} & 1 \end{array}\right]\left[\begin{array}{c} \begin{array}{c} x\\ y \end{array}\\ 1 \end{array}\right] \end{array}$$


As explained in Section 4.1, when stitching a sequence of images, {H} are multiplied in series to obtain $\left\{H^{*}\right\}$. This method has the advantages of direct and easy parallel implementation, but it also poses potential problems. If an error occurs when calculating $H_{i}$, all of the following $\left\{H_{i}^{*},H_{i+1}^{*},\ldots \ldots ,H_{n}^{*}\right\}$ will be affected, which will destroy the stitching result. We propose a method to prevent this instance.

The change of perspective will generally cause 4 kinds of image transformation: displacement transformation T, scale transformation K, rotation transformation R, and perspective transformation P. The transformation matrix can be written as a combination of these four transformations:
(14)$$\begin{array}{c} H=TKRP=\left[\begin{array}{cc} \begin{array}{cc} 1 & 0\\ 0 & 1 \end{array} & \begin{array}{c} t_{1}\\ t_{2} \end{array}\\ \begin{array}{cc} 0 & 0 \end{array} & 1 \end{array}\right]\left[\begin{array}{cc} \begin{array}{cc} k & 0\\ 0 & k \end{array} & \begin{array}{c} 0\\ 0 \end{array}\\ \begin{array}{cc} 0 & 0 \end{array} & 1 \end{array}\right]\left[\begin{array}{cc} \begin{array}{cc} cos\theta & -sin\theta \\ sin\theta & cos\theta \end{array} & \begin{array}{c} 0\\ 0 \end{array}\\ \begin{array}{cc} 0 & 0 \end{array} & 1 \end{array}\right]\left[\begin{array}{cc} \begin{array}{cc} 1 & 0\\ 0 & 1 \end{array} & \begin{array}{c} 0\\ 0 \end{array}\\ \begin{array}{cc} p_{1} & p_{2} \end{array} & 1 \end{array}\right] \end{array}$$
(15)$$\begin{array}{c} H=\left[\begin{array}{ccc} kcos\theta & -ksin\theta & t_{1}\\ ksin\theta & kcos\theta & t_{2}\\ p_{1} & p_{2} & 1 \end{array}\right] \end{array}$$

Denote the set {H|H satisfies Equation (12)} as ${ℋ}_{1}$, {H|H satisfies Equation (15)} as ${ℋ}_{2}$. We will find that ${ℋ}_{2}-{ℋ}_{1}\neq \emptyset$. Because H in ${ℋ}_{1}$ has 8 degree of freedom, while H in ${ℋ}_{2}$ has 6. That means some of H that we find according to assumption (12) and Equation (13) are not reasonable. Based on this conclusion, we add two constraints to ensure that H is reasonable, that is:
(16)$$\begin{array}{c} \{\begin{array}{c} h_{11}=h_{22}\\ h_{12}=-h_{21} \end{array} \end{array}$$

In practice, when we obtain a transformation matrix H, we test it with these constrains. If they are approximately satisfied, which means
(17)$$\begin{array}{c} \{\begin{array}{c} \left|h_{11}-h_{22}\right|<\varepsilon \\ \left|h_{12}+h_{21}\right|<\varepsilon \end{array} \end{array}$$where ε is the threshold, then we judge that H is reasonable. Otherwise we discard this H and use the result of rough registration as the positional relationship of the two pictures.

By using this method, we can generally prevent overall stitching failure due to a single stitching error between two images in the image sequence.

### 4.5. The Specific Process of Stitching Two Adjacent Images

According to Section 4.2, Section 4.3 and Section 4.4, the process of matching two adjacent images, I1 and I2, is determined. First, an anchor is selected from feature points in I1 according to the principles described in Section 4.2. Second, this anchor is matched according to the method described in Section 4.3. If this feature point pair meets the requirements in Section 4.3.3, this point pair is recorded as an anchor point pair, otherwise the anchor point is discarded and we continue to find the next anchor point. We repeat these two steps until one anchor pair is found in each grid described in Section 4.2.3, part c). Finally, according to the method described in Section 4.4, the transform matrix is calculated by means of the anchor point pairs. The process is detailed in Algorithm 1.
**Algorithm 1:** The Pseudo-Code of Stitching Two Adjacent ImagesInput: Adjacent images I1, I2;The camera parameter of images camera1, camera21. Use camera1, camera2 to get the rough registration of two images f2. Use S-H algorithm [34] and rough registration to get the overlapped area of the two images3. Use SURF to extract feature points {pi} (i=1…m) in the overlapped area on I14. Dived the overlapped area into n×n grids, and AreaFlagi← FALSE (i=1,…,n2)5. Sort feature points, such that pi.response>pi+1.response (i=1,…,m)6. AnchorKeypointPairs←{}7. **for**
pi
**in**
{pi}:8. **if** (pi is in the kth grid) **and** (AreaFlagk=FALSE):9.   pi is used as anchor point, its corresponding position: (u,v)←f(pi.x,pi.y)10.   let region R be the neighbor window around (u,v) on I2 with margin=3σ+pi.FeatureSize11.   extract SURF features points in R →
{qj}(j=1…l)12.   **for**
qj
**in**
{qj}:13.   Δj=|pi.descriptor−qj.descriptor|14.   **if**
Δmin<threshold:15.    push 〈pi,qmin〉 into set AnchorKeypointPairs16.    AreaFlagk←TRUE17.   **if** all the AreaFlag=TRUE:18.   break19. Solve transform matrix H with AnchorKeypointPairs20. **if**
|h11−h22|>ε or |h12+h21|>ε (hij is the term of H):21.  H← use rough registration f(x,y) to calculate transform matrix22. Use H to stitch the images to get the stitched imageOutput: a stitched image I

### 4.6. Theoretical Analysis of Computational Complexity

To analyze the reduction of computation in optimization strategies presented in Section 4.2 and Section 4.3, we assume the size of $I_{1}$ and $I_{2}$ is M×N and the average density of the feature point is one feature point per P pixel. The overlap rate is r. The size of the neighbor window is 2 m×2 m, and the num of grids is n×n. The possibility that the found match does not meet the threshold mentioned in Section 4.3.3 is $p_{\Delta }$. Then, in the traditional whole image match, the number of feature points extracted and descripted from two images $I_{1}$ and $I_{2}$ is:
(18)$$\begin{array}{c} p_{1}=p_{2}=\frac{M\times N\;}{P} \end{array}$$
while in FUIS, the number of feature points extracted from $I_{1}$ is:
(19)$$\begin{array}{c} p_{1}^{’}=r\frac{M\times N\;}{P} \end{array}$$and the number of feature points extracted from $I_{2}$ is:
(20)$$\begin{array}{c} p_{2}^{’}=n\times n\times \frac{1}{1-p_{\Delta }}\frac{\;2m\times 2m\;}{P} \end{array}$$

In the traditional method, the number of feature-matches we need to make is:
(21)$$\begin{array}{c} q=p_{1}p_{2} \end{array}$$while in our method, we only need to match the feature points in $I_{1}$ with the points in its corresponding neighbor window in $I_{2}$. The number of matches is:
(22)$$\begin{array}{c} q^{\prime }=p_{2}^{’}=n\times n\times \frac{1}{1-p_{\Delta }}\frac{\;2m\times 2m\;}{P} \end{array}$$

In conclusion, using our strategies, the amount of feature extraction and description is reduced to $\frac{r}{2}+\frac{4n^{2}m^{2}}{MN\left(1-p_{\Delta }\right)}$ of the traditional approach, and the time of required for feature matching is reduced to $\frac{4n^{2}m^{2}}{M^{2}N^{2}\left(1-p_{\Delta }\right)}$ of the traditional approach.

## 5. Experiments and Analysis

### 5.1. Experimental Settings

To verify the performance of the proposed FUIS approach, several experiments were carried out on two sets of images data:

Data set 1. A sequence consisting of 20 UAV images of Xishuangbanna, Yunnan Province, China. They were taken by DJI PHANTOM 3, and are 4000×3000 pixels each. The features in the image are mainly houses, so the feature points are relatively easy to extract. However, houses also might lead to parallax in the stitching results. The average overlap rate of adjacent images is 85.1075%.

Data set 2. A sequence consisting of 15 UAV images of Changed, Hunan province, China. They are provided by Beijing Yingce Space Information Yechnology Co. LTD. Their size is 7952×5304. The ground features in the images are mainly farmland. The ground is flat, but it is difficult to extract significant features. The average overlap rate of adjacent images is 71.9428%.

Figure 3 displays these two data sets. These images use EXIF (exchangeable image file format) to record their position and attitude information.

The following experiments are completed on a laptop computer with an Intel^®^ Core™ I7-8750H CPU, 2.20GHz, and 16G RAM. Programming tools and development platforms are Visual Studio 2019 and OpenCV 2.4.1. 

In this paper, we select SURF [11], ORB [13], and GMS [19] as baselines. SURF and ORB are mainstream methods for feature extraction and description. 

SURF (speeded up robust features) [11] applies the pyramid of DoG (difference of Gaussian) image and orientation assignment, thus providing a great degree of rotation and scale invariance. It is similar to SIFT but faster because of it utilizes acceleration methods, box filter and Haar filter [5]. Therefore, SURF is usually used in UAV image stitching [3,5]. The parameters of SURF are set as follows: set the threshold for the hessian keypoint detector to 100. Use 64-element descriptors rather than 128-element. Set the number of pyramid octaves to 4. Set the number of octave layers within each octave to 3. 

ORB (oriented FAST and rotated BRIEF) [13] is a combination of two excellent feature-point algorithms: FAST feature point extractor and BRIEF feature point descriptor, and it achieves excellent results in UAV image stitching [4]. Using these two feature points, we use existing methods, whole image match, as our baseline. When extracting ORB feature points, too many feature points will lead to numerous computations, while too few will lead to inaccuracy in stitching. Based on the experiment, we set the maximum number of features to 100000.

GMS (grid-based motion statistics) [19] is a novel approach that uses ORB as the feature descriptor, and selects the correct matching feature point by considering the relative positional relationship between the feature point pairs, thus significantly improving the accuracy of feature point matches. We use default parameters when applying GMS.

The baselines and some steps of FUIS, including feature extraction and description, transformation matrices calculation, and seams extraction are partially based on the implementation of OpenCV [35].

### 5.2. Experimental Analysis of Computational Complexity

To verify the computational analysis of Section 4.6, we selected a pair of adjacent images from each of the two image sequences described in Section 5.1. We stitched these two image pairs with FUIS, SURF, ORB, and GMS, respectively, and analyzed the computational complexity. The result is recorded in Table 1.

In Table 1, in addition to extracting feature points and matching, the total time also includes the time required to calculate H and other calculations. The matching time of FUIS includes the time wasted on extracting feature points in the neighborhood window and feature point matching.

As can be seen from the table, as previously analyzed, the feature extraction time can be reduced to less than half of the traditional SURF. OpenCV has optimized the matching algorithm by establishing indexes and applying other methods. However, due to the large number of feature points, the matching time remains extensive, while FUIS significantly reduces the matching time. This advantage is more obvious when the image size is larger and the number of feature points is greater. These experimental results prove the theoretical analysis in Section 4.6, that FUIS can greatly reduce the calculations for both extracting and matching.

### 5.3. Results Comparison

We apply these approaches to stitch the UAV images and evaluate their performance based on computing time and accuracy.

#### 5.3.1. Computing Time

We stitched the image sequence by FUIS, SURF, ORB, and GMS, respectively, and recorded the matching results and the time they used. The time is recorded in Table 2, and Figure 4 presents the matching results of FUIS.

The experiment results above indicate that FUIS is faster than any mainstream approach. Compared to traditional SURF, the time cost by FUIS on the two datasets was reduced by 69.63% and 72.74%, respectively. When parallel processing was applied, the reduction in time did not achieve the expected increase. Because the SURF of OpenCV [35] that we used is already been optimized with parallel processing. Therefore, even without our parallel processing method, the computing power of our device was almost fully utilized.

#### 5.3.2. Accuracy

The purpose of the accuracy experiment is to find out that to what extent FUIS improves the accuracy of rough registration, and whether FUIS can achieve an accuracy comparable to that of mainstream non-simplified approaches. To this end, we picked two adjacent images from each of the two data sets to compare the stitching result of the rough registration result, FUIS and the baseline mentioned above.

To quantize the error, according to the definition of the stitching problem in Section 3, we manually select a set of corner point pairs in the two adjacent images, as displayed in Figure 5. The points are denoted as $P=\left\{p_{1},p_{2,}\ldots ,p_{n}\right\}$ and $P^{\prime }=\left\{p_{1}^{’},p_{2}^{’}\ldots ,p_{n}^{’}\right\}$. We then calculate the mean deviation of corner points after stitching. Figure 6 demonstrates the stitching result of FUIS with corner points on it.
(23)$$\begin{array}{c} b_{f}=\frac{1}{n}\sum \left|f\left(p_{i}\right)-p_{i}^{’}\right|\;,\;p_{i}\in P,p_{i}^{’}\in P^{’} \end{array}$$where f is the image transformation function to be tested.

As can be seen from Table 3, the error of FUIS is comparable to mainstream approaches. Comparing the results of rough registration, the error is much larger than the other approaches. This indicates that FUIS significantly reduces the error of rough registration, and achieves the accuracy of mainstream homography-matrix-based stitching models. 

### 5.4. The Influence of the Parameters in FUIS

#### 5.4.1. Grid Density

To find a proper grid density for feature reduction and anchor point selection, we use the algorithm with different grid rows and columns to stitch the 2 adjacent images, record the time used, and evaluate the stitching accuracy with the mean deviation defined in 5.3.2.

Figure 7 indicates that when the grid density is less than 4×4, the stitching accuracy decreases significantly. If it is larger than 4×4, the accuracy does not continue to drop dramatically, but there is a corresponding rise in the amount of time used. Therefore, in FUIS, a 4×4 grid is used.

#### 5.4.2. The Threshold of Δ

To ensure that the feature matches are correct, we conducted an experiment to find an appropriate threshold for Δ (which is the Euclidean distance between the feature descriptors mentioned in 4.3.3).

With the data sets mentioned in Section 5.1, we obtained 260 feature matches picked by FUIS, manually judged whether the matching is correct, and recorded the Δ of matches. Among them, there were 107 incorrect matches and 153 correct matches. We set the incorrect match as “Positive” and drew a receiver operating characteristic (ROC) figure based on these data. The ROC figure is shown in Figure 8.

The AUC (area under the curve) is 0.9483, which means that using Δ to judge the correctness is reasonable. According to the ROC figure, we set the threshold at 0.042516, where the TPR (the probability of correctly removing the match when it is an incorrect match) is 95%. It should be noted that this threshold is only applicable to FUIS, because the error matching point here is not an arbitrary error match, but an error match that passes the various filters mentioned in Section 4.3. It is in the neighbor window and has a similar feature scale.

## 6. Conclusions

For the requirement of emergency stitching of UAV images, this paper proposes FUIS that can significantly decrease the computation amount of stitching with the use of position and pose information. This approach meets the emergency requirement efficiently and quickly. This benefit results from the use of the proposed anchor point selection and matching method. According to the experiment, our stitching accuracy is not lower than the mainstream methods, while the speed has been greatly improved.

Since in order to improve speed, we use image transformation instead of orthomosaic generation to stitch the images, our method may have errors when stitching images of areas with large terrain undulations. 

## Figures and Tables

**Figure 1 sensors-20-02007-f001:**
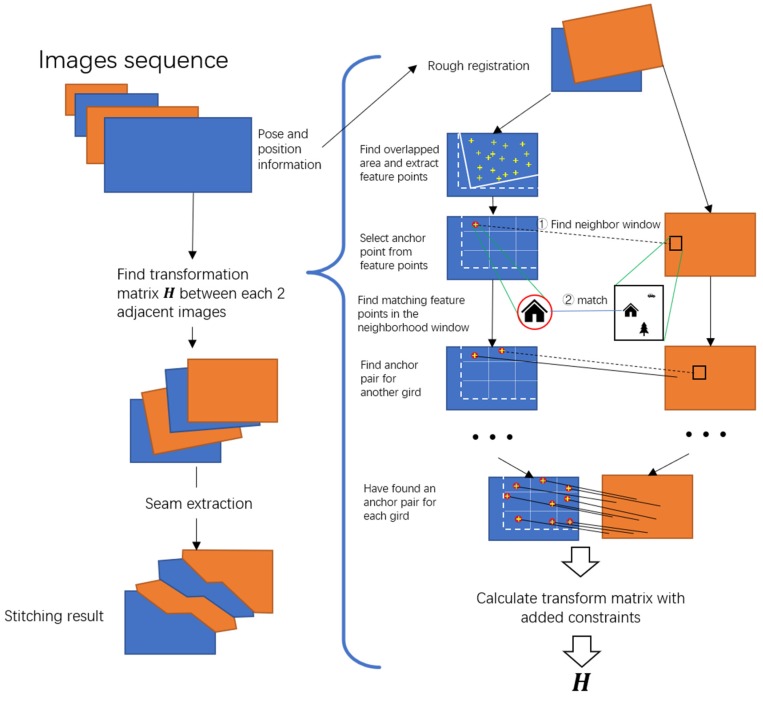
Flow diagram of fast unmanned aerial vehicle (UAV) image-stitching (FUIS) program. The right side is the process of finding the transformation matrix between two adjacent images, and the left side is the process of stitching an image sequence.

**Figure 2 sensors-20-02007-f002:**
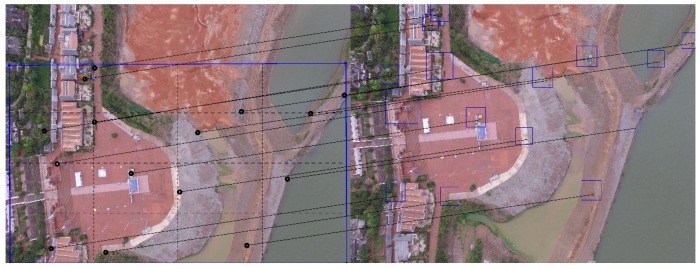
Find the matching feature points in the neighborhood.

**Figure 3 sensors-20-02007-f003:**
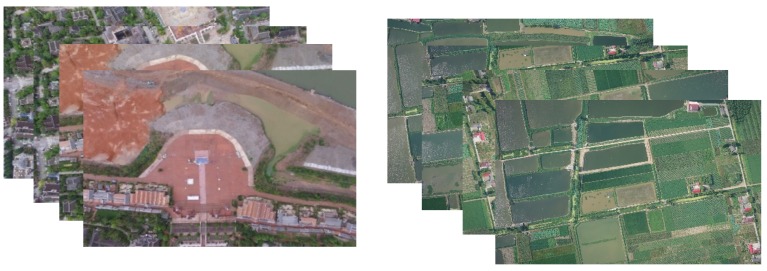
Two image sequences, (only four of which are shown here. The **left** one is data set 1. The **right** one is data set 2.).

**Figure 4 sensors-20-02007-f004:**
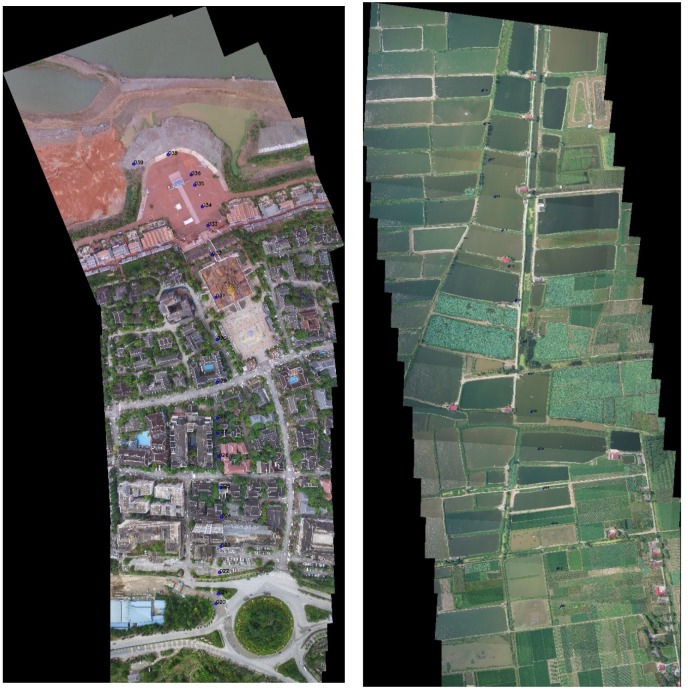
Stitching result of image sequences (the **left** one is the result of data set 1. The **right** one is data set 2.).

**Figure 5 sensors-20-02007-f005:**
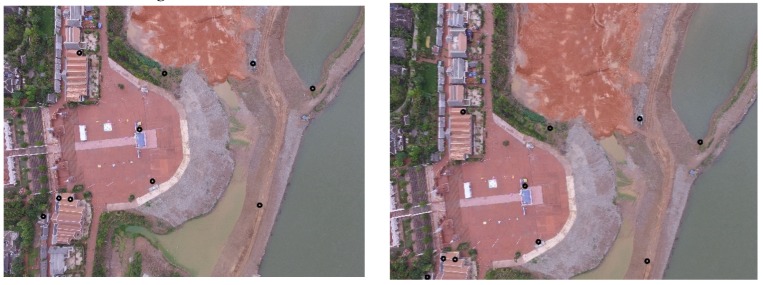
Manually selected feature points in two images.

**Figure 6 sensors-20-02007-f006:**
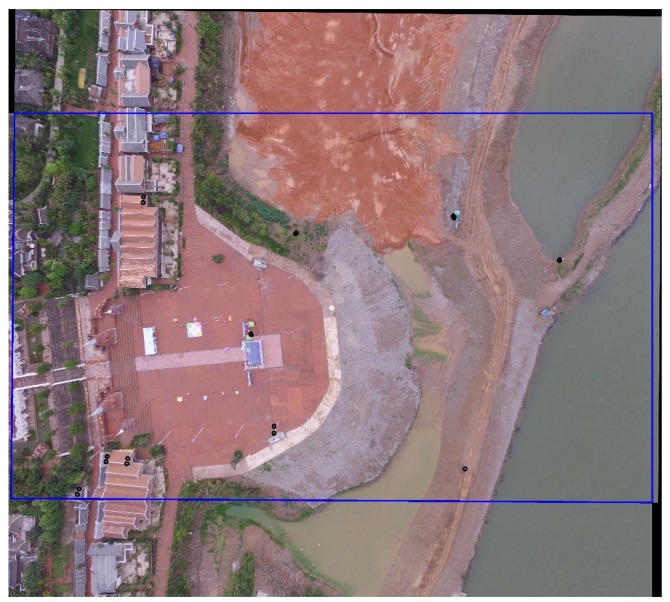
The stitching result of FUIS with corner points on it.

**Figure 7 sensors-20-02007-f007:**
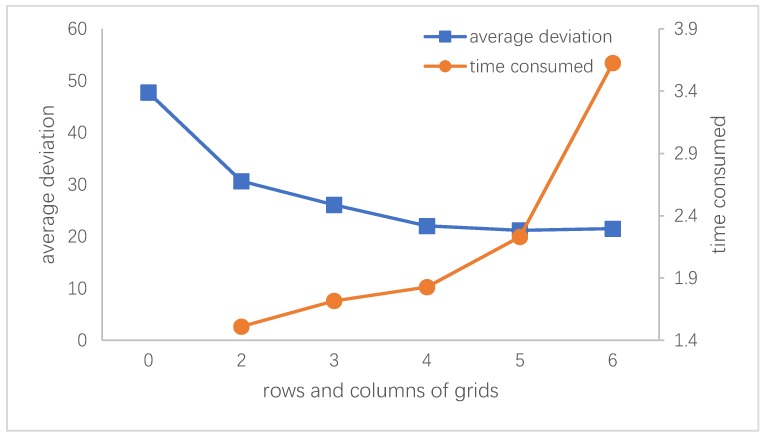
The influence of grid number. 0 grids represent the result of rough registration.

**Figure 8 sensors-20-02007-f008:**
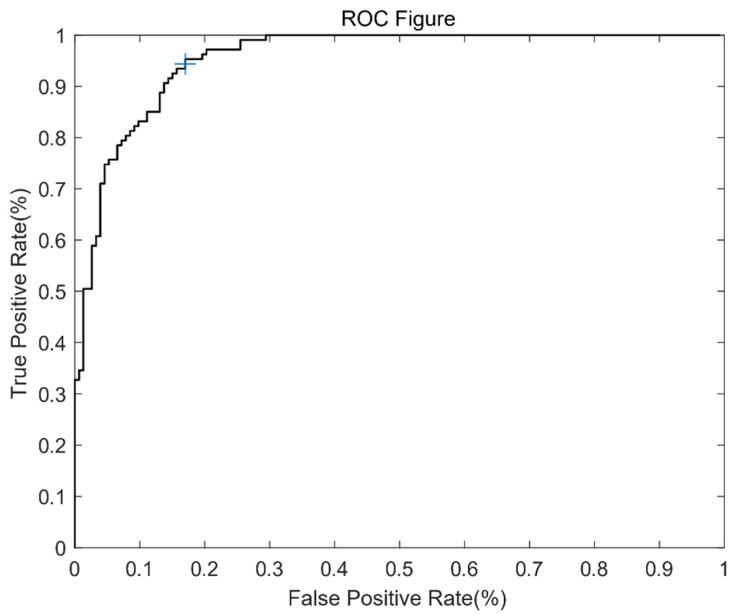
The receiver operating characteristic (ROC) figure for Δ. The + sign denotes the location of the selected threshold.

**Table 1 sensors-20-02007-t001:** Computational complexity of different approaches.

4000×3000 Images	**Time(Second)**			**Number of Feature Points**
	**Total**	**Extracting**	**Matching**	
**FUIS**	1.763	1.448	0.305	Checked 2010 anchor points
**SURF**	5.148	3.917	1.214	58,264×58,745
**ORB**	33.733	2.110	32.255	91,391×91,491
**GMS**	44.543			
7952×5304 **Images**	**Time(Second)**			**Number of Feature Points**
	**Total**	**Extracting**	**Matching**	
**FUIS**	12.457	6.697	5.746	Checked 593 anchor points
**SURF**	58.047	20.046	37.969	351,873×334,461
**ORB**	49.161	3.384	38.436	100 K × 100 K
**GMS**	65.457			

**Table 2 sensors-20-02007-t002:** Time cost by the methods to stitch the image sequence (second).

	FUIS	FUIS Using Parallel Processing	SURF	ORB	GMS
**Data set 1**	74.234	62.113	244.392	447.952	448.307
**Data set 2**	176.964	146.747	649.242	685.805	861.097

**Table 3 sensors-20-02007-t003:** Mean deviation of different methods’ stitching result (pixel).

	Rough Registration	ORB	GMS	SURF	FUIS
**Data set 1**	47.73561	17.86671	17.74659	21.07692	22.04122
**Data set 2**	243.3556	12.9921	5.85734	4.270076	6.284166

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
