# Peer review of "Fast Anchor Point Matching for Emergency UAV Image Stitching Using Position and Pose Information"

_sensors, 2020, doi:10.3390/s20072007_

Round 1
Reviewer 1 Report
The paper presents a novel method to stitch images with possible applications in UAVs. Authors claim that comparisons have been achieved with respect a state-of-art method having better results. The following observations are in order:
- Compared to other methods, what is novelty in the selection of the feature points described in Section 4.
- With respect to results published in
“Fast image stitching of unmanned aerial vehicle remote sensing image based on SURF algorithm”, Man Yuan; Tianjie Lei; Xuemei Liu; Shican Li, Proceedings Volume 11179, Eleventh International Conference on Digital Image Processing (ICDIP 2019); 111791A (2019) https://doi.org/10.1117/12.2548647
What are the main differences of the intended method? The above-cited manuscript presents a similar comparison of methods. I wonder if authors may include that algorithm in the comparison with respect the novel method.
- Image stitching a very large field. Paper presents only 4 references published in 2018 and 2019. Authors should demonstrate through an appropriate literature review that the topic is relevant by adding more references.
- I wonder if the method can be sped up to achieve some real-time processing and, if possible, how could it be applied in visual servoing in order an UAV follows an object.
Reviewer 2 Report
The work is interesting and covers a topic addressed by the Journal. However, the paper needs further revision in order to better clarify some aspects of both the algorithm developed and the writing of the manuscript.
In the section 5.1., how much is the overlap value between the images? This parameter should affect your algorithm?
The section 4.5 needs to be better described.
The figures 3, 4, 5 and 6 must be called in the manuscript.
The tables 1 and 2 must be called in the manuscript.
I suggest to evaluate better this sentence "In future work, we consider using the DEM map to correct the result,
which is might able to improve the accuracy without adding too much computation"
What software was used for the experiments?
Minor revision:
The tables and some references must be formatted according the indication of the Journal: https://www.mdpi.com/journal/sensors/instructions#preparation
I suggest to delete the reference in the title of the section 4.2.2.
Often the acronym "UAV" has been misspelled in the paper and indicated with "UVA".
Specify the following acronyms: GPS, SIFT, SURF, RANSAC, etc.
Round 2
Reviewer 1 Report
The proposed method for images stitching have been tested and compared with different techniques.
The references have been updated.
I found very interesting the answer on the possible application in a real-time implementation.
The authors have answered satisfactorily my questions. The recommend publication of this manuscript in its present form.
Reviewer 2 Report
I have no further comments